# Glassy and Rubbery Epoxy Composites with Mesoporous Silica

**Dimitrios Gkiliopoulos** [1,2,*], **Dimitrios Bikiaris** [3], **Doukas Efstathiadis** [4] and **Konstantinos Triantafyllidis** [1,2,*]

1   Laboratory of Chemical and Environmental Technology, Department of Chemistry, Aristotle University of Thessaloniki, GR-54124 Thessaloniki, Greece
2   Center for Interdisciplinary Research and Innovation (CIRI-AUTH), Balkan Center, 10th km Thessaloniki-Thermi Rd., GR-57001 Thessaloniki, Greece
3   Laboratory of Polymers and Colors Chemistry and Technology, Department of Chemistry, Aristotle University of Thessaloniki, GR-54124 Thessaloniki, Greece; dbic@chem.auth.gr
4   Stone Group International, Kavalari, GR-57200 Thessaloniki, Greece; efstathiadis.d@stonegroup.gr
*   Correspondence: dgiliopo@chem.auth.gr (D.G.); ktrianta@chem.auth.gr (K.T.)

**Abstract:** The reinforcing efficiency of SBA-15-type mesoporous silica, when used as additive in epoxy polymers, was evaluated in this study. The effects of silica loading and its physicochemical characteristics on the thermal, mechanical, and viscoelastic properties of glassy and rubbery epoxy mesocomposites were examined using SBA-15 mesoporous silicas with varying porosities (surface area, pore size, and volume), particle sizes, morphologies, and organo-functionalization. Three types of SBA-15 were used: SBA-15 (10) with 10 nm pore diameters and long particles, SBA-15 (5) with 5 nm pore diameters and short particles, and SBA-15 (sc) with 10 nm pore diameters and short particles ("sc" for short channel). SBA-15 (10) was modified with propyl-, epoxy-, and amino-groups to study the effect of functionalization. The glassy or rubbery epoxy polymers and mesocomposites were produced by the crosslinking of a diglycidyl ether of bisphenol A (DEGBA) epoxy resin with isophorone diamine (IPD) or Jeffaminje D-2000, respectively. Mesoporous silica was uniformly dispersed inside the polymer matrices; however, the opacity levels between the rubbery and glassy samples were different, with completely transparent rubbery composites being prepared with as high as a 9 wt. % addition of SBA-15. The mechanical and thermal performance properties of the mesocomposites were dependent on both the type of the curing agent, which affected the cross-linking density of the pristine polymer matrix, and the characteristics of the mesoporous silica variants, being, in general, improved by the addition of up to 6 wt. % silica for the glassy polymers and up to 9 wt. % for the rubbery polymers.

**Keywords:** polymer composites; epoxy; mesoporous silica; SBA-15; organo-functionalization

## 1. Introduction

Nanosized, inorganic particles have been effectively used over recent years as polymer additives to produce nano- and meso-composite materials with novel or improved properties, in comparison to pristine polymers and conventional polymer micro-composites [1–5]. The superior properties of polymer nano- and meso-composites arise from the nanoscopic particle size of their inorganic additives and the large interfacial area between their organic and inorganic phases, which allow for extended, molecular-level interactions between the polymer and its inorganic particles. For these interactions to take place, the nanoadditives must be homogeneously dispersed inside the polymer matrix in the form of primary particles [6–8]. However, nanoparticles tend to aggregate in microsized formations due to interparticle forces, resulting in composite systems with lower interfacial properties. To overcome this situation, various physical and chemical techniques are applied, which aim at the disaggregation and homogenous dispersion of these nanoparticles, such as high-shear mixing [9,10], the use of solvents [11], sonication [12], the organo-functionalization of nanoparticles [3,4,13], and others.

The most used inorganic nanoadditives used to produce polymer composites are clays [3,14], metal oxide nanoparticles [13,15,16], carbon nanotubes [8,11], and more recently, mesoporous silica [17–20], graphene [5,21–23], and nanocellulose [24–26]. Mesoporous silicas have pore sizes between 2 and 50 nm, large (>500 m$^2$/g) specific surface areas, and hydroxyl-rich surfaces. Due to their exceptional physical and chemical properties, mesoporous silicas have been rendered as ideal materials for many applications, such as catalysis [27–29], adsorption [28,30], biomedical applications [31–35], and others. Furthermore, many efforts have been conducted in recent years to utilize mesoporous silica nanoparticles as polymer-reinforcing agents based on the possible, high polymer-additive interface that can occur and the high mechanical properties of silicon dioxide (bulk modulus of 33.5–36.8 GPa, compressive strength of 1100–1600 MPa, hardness of 4500–9500 MPa, tensile strength of 45–155 MPa, and Young's modulus of 66.3–74.8 GPa) [17–20].

The types of polymers that have been used to produce mesoporous silica/polymer composites vary from epoxy resins [17,18,36,37] to acrylic polymers [19,20,38,39], polyalkanes [40,41], polyimides [42], polyamides [43], polystyrene [44], and silicone [45]. As it has been shown, the addition of mesoporous nanoparticles generally results in composite materials with higher thermal, mechanical, and thermomechanical properties, in comparison to pristine polymers, even at low concentrations [17–20,36–45]. Moreover, the in situ polymerization of various monomers in the presence of mesoporous silica has proved to be a far more effective approach to composite preparation than other mixing procedures, because of the easier penetration of small monomer molecules inside the silica pores prior to polymerization, which leads to increased interfacial interactions and improved properties in the final composites [19,39]. Regarding the pore dimensions and morphology of mesoporous silica, larger pore sizes and 3D structured frameworks correspond to nanocomposites with increased thermomechanical properties [20]. Furthermore, the tensile properties of nanocomposites are mostly dependent on the volume of the polymer that impregnates the silica mesopores, rather than the interfacial interactions, indicating a higher impact of the mesopores' volume on the mechanical properties over the silica specific surface [17].

In the present work, both glassy (plastic) and rubbery (elastomeric) epoxy/mesoporous silica composites were prepared, using various types of SBA-15 mesoporous silicas as nanoadditives. A cycloaliphatic amine (IPD) and high-molecular-weight polyetheramine (Jeffamine D-2000) were used as curing agents to produce the glassy and rubbery polymer matrices, respectively. The aim of this study was to investigate the effects of the mesoporous silicas' morphologies, porosities, organophilicities, and surface activities on the properties of the prepared epoxy composites.

## 2. Materials and Methods

### 2.1. Materials

Tetraethyl orthosilicate (TEOS) was purchased from Aldrich and used as the silica precursor of the mesoporous silica. Triblock copolymer Pluronic® P123 (EO$_{20}$PO$_{70}$EO$_{20}$, MW 5800, BASF, Ludwigshafen, Germany) was the structure-directing agent. The surface modification of the SBA-15 particles with propyl-, epoxy-, and amino-groups was conducted using propyl triethoxysilane (PTES, Sigma-Aldrich, St. Louis, MO, USA), 3-glycidyloxypropyl triethoxysilane (GPTES, Aldrich), and 3-aminopropyl triethoxysilane (APTES, Aldrich, Burlington, MA, USA), respectively. EPON 827 epoxy resin (Huntsman, The Woodlands, TX, USA) was crosslinked with poly(propylene oxide)-α,ω-diamine Jeffamine D-2000 (Huntsman) and isophorone diamine (IPD, Sigma-Aldrich, St. Louis, MO, USA) to prepare the rubbery and glassy epoxy polymers, respectively.

### 2.2. Synthesis of SBA-15 Mesoporous Silicas

SBA-15 mesoporous silicas with different pore and particle sizes were prepared according to previous methodologies [46] via acid catalyzed sol-gel reactions, following the cooperative self-assembly route. Three variants were synthesized in total. SBA-15

(10) with 10 nm pore diameters, SBA-15 (5) with 5 nm pore diameters, and SBA-15 (sc) with 10 nm pore diameters and shorter pore (channel) lengths. In this typical synthesis, Pluronic® P123 was added to aqueous HCl 1.6 M and stirred until the formation of a crystal-clear solution. TEOS was then added drop wise and the mixture was stirred under different conditions for each SBA-15 type, followed by a hydrothermal treatment in a polypropylene autoclave. The filtrated products were washed with deionized water and ethanol, dried at an ambient temperature, and finally calcined in air to remove their organic content. The molar compositions for all the syntheses were the same and equal to TEOS (1):P123 (0.018):HCl (3.33):$H_2O$ (97.5). The specific synthesis conditions for each SBA-15 variant are presented in Table 1.

**Table 1.** Synthesis procedures of different SBA-15 mesoporous silicas.

| Synthesis Step | SBA-15 (10) | SBA-15 (5) | SBA-15 (sc) |
|---|---|---|---|
| Pluronic® P123 solubilization in HCl$_{(aq)}$ 1.6 M | Stirring at 38 °C until dissolved | Stirring at 38 °C until dissolved | Stirring at 38 °C until dissolved |
| TEOS hydrolysis and polymerization | Stirring at 38 °C for 24 h | Stirring at 35 °C for 1 h | Stirring at 40 °C for 8 min |
| Hydrothermal treatment | 100 °C, 72 h | 35 °C, 48 h | 40 °C, 24 h |
| Product recovery | Filtration. Wash with deionized water and ethanol. Drying at T$_{room}$ | Filtration. Wash with deionized water and ethanol. Drying at T$_{room}$ | Filtration. Wash with deionized water and ethanol. Drying at T$_{room}$ |
| Organic content removal | Calcination at 550 °C for 6 h with 1 °C·min$^{-1}$ | Calcination at 550 °C for 6 h with 1 °C·min$^{-1}$ | Calcination at 550 °C for 6 h with 1 °C·min$^{-1}$ |

### 2.3. Surface Functionalization of SBA-15 Mesoporous Silicas

The surface functionalization of SBA-15 (10) with propyl-, epoxy-, and amino-groups was accomplished via a post-synthesis treatment of the calcined silica with PTES, GPTES, and APTES, respectively, in anhydrous toluene purged with nitrogen. In this typical experiment, 2 g of SBA-15 (10) was added to 66.67 mL (0.63 mol) of toluene, which had previously been degassed with nitrogen for 2 h. After the silica dispersion, 0.113 mol of organic modifier was added and the mixture was allowed to react at 60 °C for 24 h. The functionalized silica was recovered with filtration, washed with toluene (1 wash), ethanol (2 washes), and deionized water (2.5–3 L), and was allowed to dry at room temperature. A schematic representation of the SBA-15 surface functionalization is given in Scheme 1.

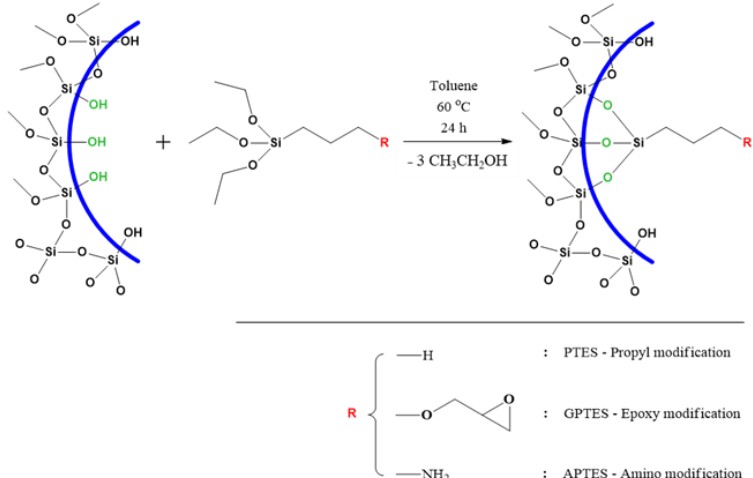

**Scheme 1.** Surface functionalization of SBA-15 pores with alkyl-triethoxy silanes.

### 2.4. Preparation of Pristine Epoxy Polymers and SBA-15 Mesocomposites

Pristine epoxy polymers were prepared via a ring-opening polymerization of the epoxy resin with diamine curing agents. The cross-linking of each (epoxy resin/curing agent) system was accomplished using Teflon molds to form standard-shaped specimens. In a typical procedure, the epoxy resin was heated at 50 °C under stirring to lower the viscosity and remove air bubbles, followed by a curing agent addition and stirring for another 10 min. The mixture was then outgassed in a vacuum oven at 50 °C for 10 min, prior to pouring in the molds. Each system was then cured under specific conditions. The rubbery epoxy polymers were prepared after heating DEGBA and Jeffamine D-2000 (per hundred resin, phr 283.20 g) at 60 °C for 3 h (first-step curing) and at 125 °C for 3 h (second-step curing). The glassy epoxy polymers were prepared by heating DEGBA and IPD (phr 23.46 g) at 25 °C for 24 h (first-step curing) and at 160 °C for 2 h (second-step curing).

The epoxy/SBA-15 mesocomposites were prepared via in situ polymerization. The mesoporous silicas were dispersed in the already heated epoxy resin for 1 h prior to the addition of a curing agent, while the rest of the process (outgassing and curing) was conducted by following the same standard procedure for each system.

### 2.5. Measurements

Scanning electron microscopy (SEM) was performed using a JEOL JMS 7610 F (Jeol, Freising, Germany) scanning electron microscope, operating at 10 kV and equipped with an energy dispersive X-ray (EDX) Oxford ISIS 300 micro-analytical system. The specimens were prepared by placing SBA-15 silica powder on the SEM holder, followed by gentle shaking to remove the easily detachable dust/powder.

High-resolution transmission electron microscopy (HRTEM) was performed using a JEOL 2011 high resolution transmission electron microscope with an LaB6 filament, an accelerating voltage of 200 kV, a point resolution of 0.23 nm, and a spherical aberration coefficient of Cs = 1 mm. The samples were placed onto a carbon lacey film supported on a 3 mm diameter and 300 mesh copper grid. The specimens were further coated with a carbon layer in order to enhance their conductivity.

The $N_2$ adsorption/desorption isotherms of the mesoporous silicas were obtained at $-196$ °C on an Automatic Volumetric Sorption Analyzer (Autosorb-1MP, Quantachrome Instruments, Boynton Beach, FL, USA). The samples were outgassed at 150 °C and $1.33 \times 10^{-4}$ Pa for a minimum of 12 h prior to analysis. The surface areas were calculated using the Brunauer–Emmett–Teller (BET) analysis method and the pore size distributions were estimated using the Barrett–Joyner–Halenda (BJH) analysis model.

An elemental carbon analysis was performed using a LECO-800 CHN element analyzer to determine the number of organic groups attached on the surfaces of the modified silicas viachemical bonds. This amount can be calculated with Equation (1) below,

$$OG = w \cdot Ar_C^{-1} \cdot N^{-1} \cdot 10^{-2} \tag{1}$$

where $OG$ represents the number of the organic group moles in 1 gr of silica, $w$ is the mass fraction (wt. %) of the carbon atoms as determined by the elemental analysis, $Ar_C$ is the atomic weight of the carbon, and $N$ is the number of carbon atoms in each organic group (3 for PTES, 6 for GPTES, and 3 for APTES).

A thermal gravimetric analysis (TGA) was conducted using an SDT2956 (TA Instruments) thermobalance under a nitrogen inert gas flow (100 cc/min) and a constant heating rate of 10 °C/min in the temperature range of 25–900 °C.

The mechanical properties of the pristine epoxy polymers and SBA-15 mesocomposites were determined via tensile strength measurements using an Instron 3344 dynamometer, in accordance with ASTM D638. The crosshead speed was 50 mm/min for the rubbery materials and 5 mm/min for the glassy materials. The "dog-bone"-shaped specimens were 30 mm long in the narrow region and 2 mm thick and 5 mm wide along the center of the casting.

The resistance of the glassy materials to impact was measured using Tinius Olsen apparatus. Notched Izod impact tests were conducted according to ASTM D256. The specimens were rectangular, with dimensions of $70 \times 15 \times 2$ mm. The specimens' notches laid in the center of the long side, with a 45° angle and 2.5 mm height.

A dynamic mechanical analysis (DMA) was used to measure the thermomechanical properties using a Perkin Elmer Diamond DMA analyzer. The bending method was applied with a 1 Hz oscillation frequency and 3 °C/min heating rate. The rubbery samples were tested in the temperature range from −90 to 20 °C with a 4 mN applied force, while the glassy samples were tested in the temperature range from 25 to 180 °C with a 4000 mN applied force. The specimens had a rectangular shape with dimensions of $50 \times 13 \times 2$ mm.

## 3. Results

### 3.1. Characterization of SBA-15 Mesoporous Silicas

The morphologies of the SBA-15 mesoporous silicas were observed using SEM and TEM. As can be seen in the SEM images (Figure 1), the SBA-15 primary particles were cylindrical, apart from the SBA-15 (5) particles, which show a more cubic/parallelepiped shape. The primary particles of SBA-15 (10) were edge-agglomerated towards long rod-like particles. On the other hand, SBA-15 (5) exhibited relatively smaller aggregated clusters with wormlike or irregular shapes, while the primary particles of SBA-15 (sc) seemed to be more isolated or formed smaller aggregates of irregular shapes. However, it should be noted that these bigger formulations may well have been partially disaggregated to the smaller primary particles when dispersed in the epoxy (pre)polymer, as was also verified by the TEM images of the epoxy-silica composites discussed below.

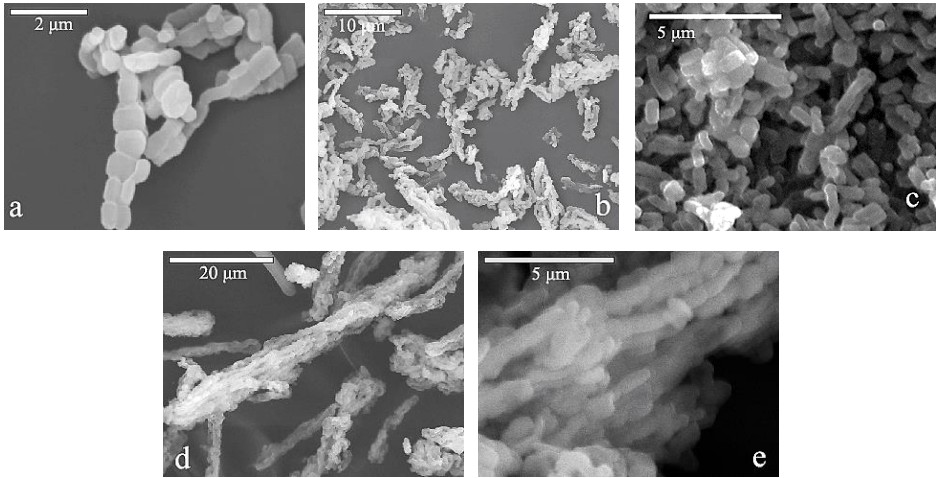

**Figure 1.** SEM images of (**a**,**b**) SBA-15 (5), (**c**) SBA-15 (sc), and (**d**,**e**) SBA-15 (10).

The TEM images of SBA-15 (10) and SBA-15 (sc) are shown in Figure 2. The tubular shapes and honeycomb-like, hexagonal arrangements of the pores are revealed by the longitudinal and transversal (inset image) cross-sections of the nanoparticles, respectively. The pore morphology of SBA-15 was attributed to the growth of the siliceous framework around the rod-like P123 micelles and, consequently, to the self-assembly of the surfactant in aqueous media. The structure of the P123 macromolecule, with the hydrophobic PPO block being surrounded by the hydrophilic PEO blocks, was responsible for the formation of cylindrical micelles [47], while the size of the micelles could be adjusted by varying the heating temperature and time [48,49]. The dimensions of the primary particles of the mesoporous silicas are shown in Table 2, as they were determined by the SEM and TEM images. SBA-15 (10) had the largest particles ($1.33 \times 0.62$ μm), while SBA-15 (5) had the shortest ($0.68 \times 0.44$ μm) and SBA-15 (sc) had the narrowest ($0.84 \times 0.30$ μm). The aspect ratios of the primary particles were: 2.15 for SBA-15 (10), 1.55 for SBA-15 (5), and 2.80 for SBA-15 (sc).

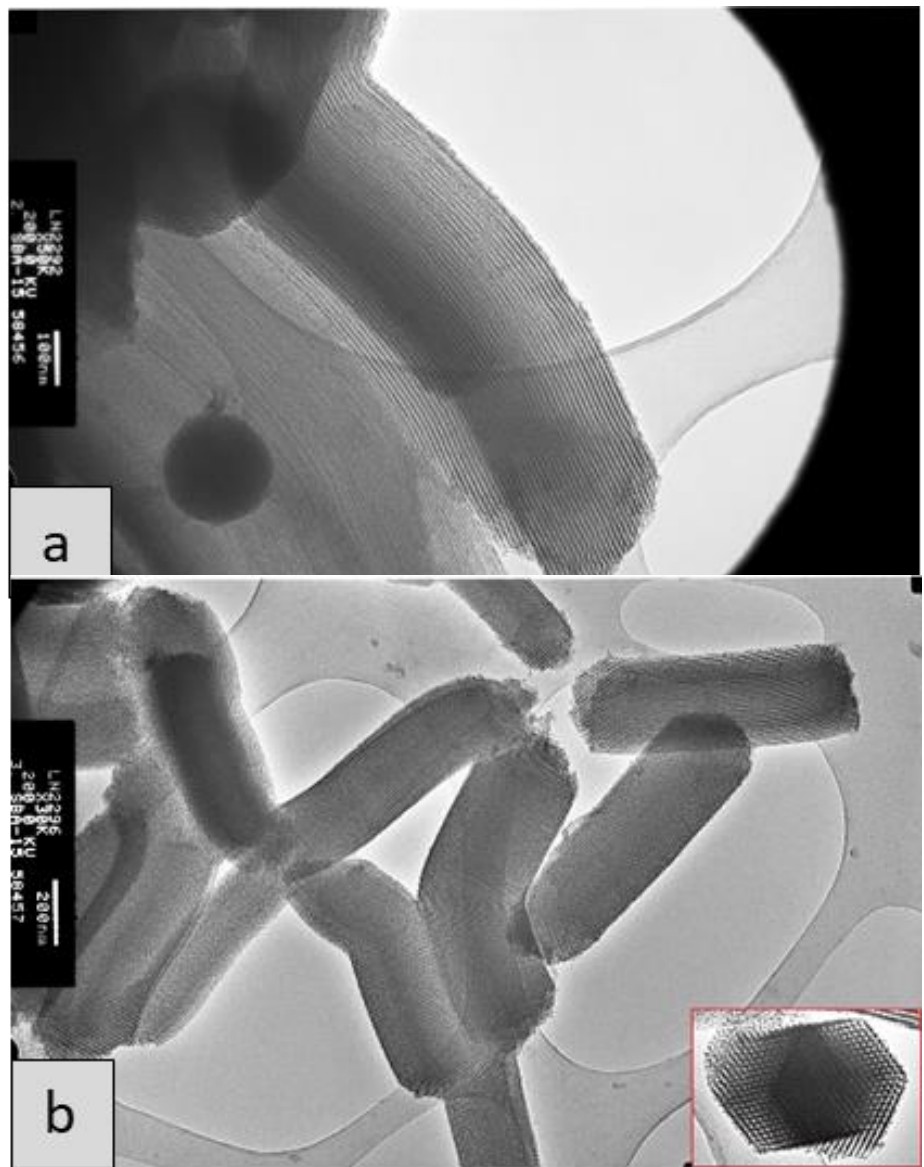

**Figure 2.** TEM image of (**a**) SBA-15 (10) and (**b**) SBA-15 (sc).

**Table 2.** Physicochemical properties of SBA-15 mesoporous silicas.

| | SEM-TEM | | N$_2$ Physisorption | | | Elemental Analysis | |
|---|---|---|---|---|---|---|---|
| Sample | Average Particle Length (µm) | Average Particle Width (µm) | Specific Surface Area (m$^2$/g) | Total Pore Volume (cc/g) | Pore Diameter (nm) | Carbon Concentration (wt. %) | Organic Group Content (mol/g) |
| SBA-15 (10) | 1.33 | 0.62 | 815 | 1.363 | 10.0 | 0 | 0 |
| SBA-15 (5) | 0.68 | 0.44 | 651 | 0.615 | 5.0 | 0 | 0 |
| SBA-15 (sc) | 0.84 | 0.30 | 780 | 1.140 | 10.0 | 0 | 0 |
| SBA-15-propyl | - | - | 735 | 1.358 | 8.9 | 1.97 | $5.5 \cdot 10^{-4}$ |
| SBA-15-epoxy | - | - | 600 | 1.251 | 8.9 | 3.47 | $4.8 \cdot 10^{-4}$ |
| SBA-15-amino | - | - | 382 | 0.758 | 8.5 | 9.54 | $26.5 \cdot 10^{-4}$ |

The effect of the synthesis conditions on the silicas' textures and porosities can be further seen from the nitrogen physisorption data in Table 2. As is evidenced, SBA-15 (5) had a smaller pore size (volume- and diameter-wise) and surface area than SBA-15 (10). Taking into account the synthesis procedures in Table 1, it can be deduced that increased

reaction times and temperatures resulted in the formation of mesoporous silicas with larger pores and surface areas, behavior that is in accordance with previous reports [46,48,49]. Another variation in the synthesis method led to SBA-15 (sc), which was characterized by a similar pore diameter to SBA-15 (10), but a shorter average particle length.

The organically modified silicas exhibited slightly smaller pore sizes and a more substantial reduction in their surface areas and pore volumes in comparison to the parent SBA-15 (10), indicating that the functionalization of the mesoporous silica was successful and that organic moieties were grafted on the surface of the mesopores. The quantitative determination of the inserted organic groups was based on the elemental (carbon) analysis results (Table 2), which also showed an interesting variation in the extent of the grafting/loading between the three organic modifiers, i.e., glycidyloxypropyl (GPTES), amino-propyl (APTES), and propyl (PTES). Although equal amounts of moles for each organic modifier were used for the functionalization of SBA-15 (10), the loading (elemental analysis) of the amino-propyl groups was substantially higher compared to the other two molecules. This was also depicted by their porous characteristics, as SBA-15-amino exhibited 53% and 44% decreases in their surface area and pore volume, respectively, compared to those of the parent silica. The respective reductions in the surface area and pore volume induced by the propyl- and glycidyloxypropyl moieties were lower, in the range of 10–26% and 0.4–8%, respectively.

### 3.2. Properties of Epoxy Polymers and Mesocomposites

Selected photographs of the pristine epoxy polymers and SBA-15 mesocomposite samples are shown in Figure 3. Epoxy polymers are amorphous materials; therefore, they are optically transparent, since there are no crystalline regions to scatter light. Concerning the optical transparency of the SBA-15 mesocomposites, different behaviors were observed between the rubbery and glassy materials. The rubbery mesocomposites, even with high loadings of mesoporous silicas or with organically modified SBA-15, were optically transparent, while the glassy mesocomposites were opaque. The opacity of the glassy mesocomposites was directly analogous to their silica content; thus, adding up to 1 % of mesoporous silica resulted in semi-transparent materials, while samples with 3 % of silica loading or higher were opaque.

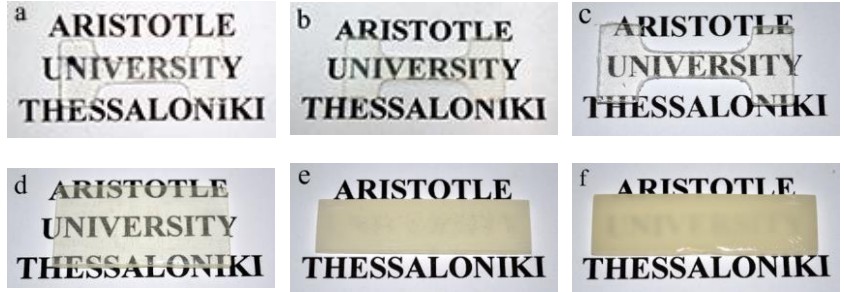

**Figure 3.** Selected photographs of SBA-15 mesocomposites: (**a**) pristine rubbery, (**b**) rubbery/9% SBA-15 (10), (**c**) rubbery/3% SBA-15-propyl, (**d**) pristine glassy, (**e**) glassy/3% SBA-15 (5), and (**f**) glassy/3% SBA-15-amino.

The dispersions and structures of the mesoporous silicas inside the polymer matrix can be directly observed in the TEM images of Figure 4. As can be seen in Figure 4a,c, the rod-like SBA-15 aggregates observed in the SEM images (Figure 1d,e) were partially disaggregated and SBA-15 was effectively dispersed inside the polymer matrix. The dispersion of mesoporous silica particles without the need for any dispersing agent has also been reported elsewhere [18]. The structures of the SBA-15 particles were not affected by the existing conditions during the mixing and curing/post-curing of the components. The well-defined hexagonal arrangement of the tubular pores of the SBA-15 particles inside the DEGBA/IPD system are visible in Figure 4b,d.

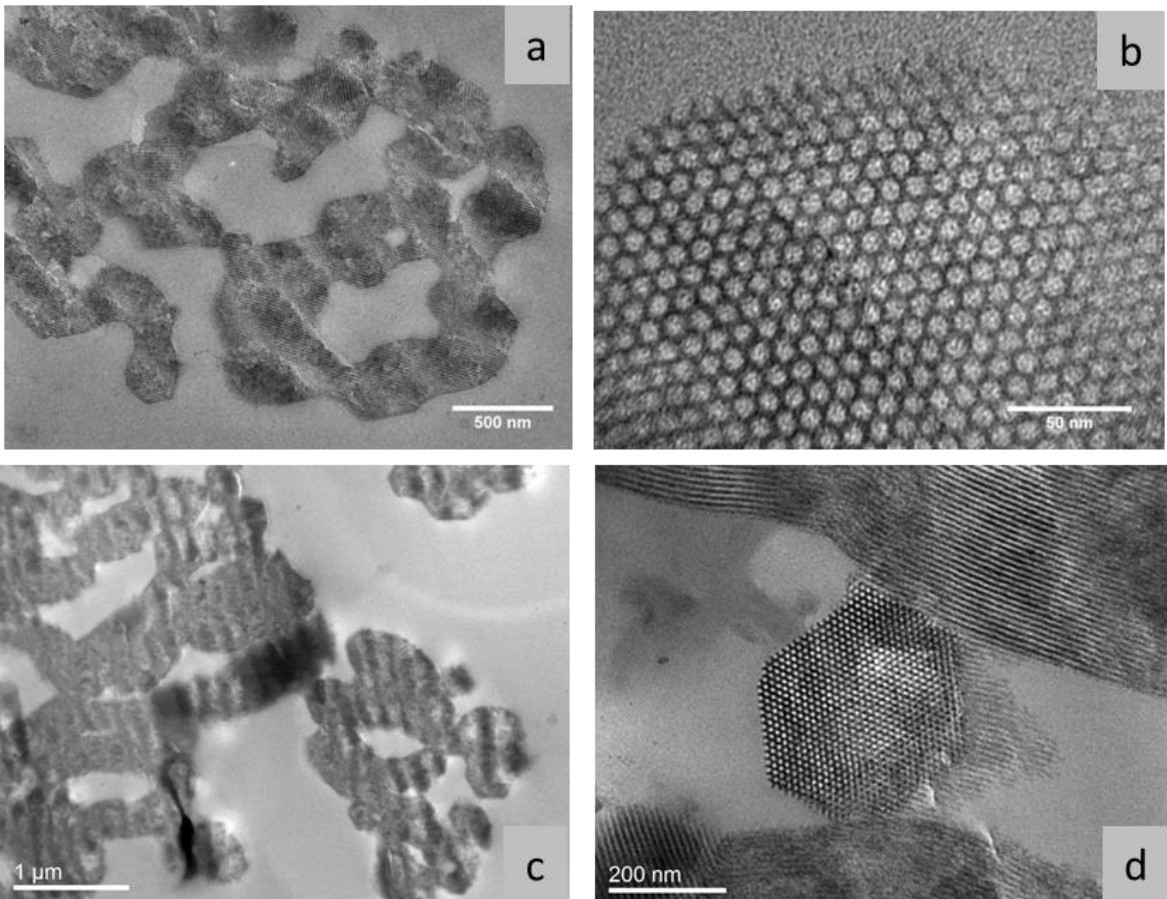

**Figure 4.** TEM images of DEGBA/IPD mesocomposites with (**a**,**b**) 3% SBA-15 (10), and (**c**,**d**) 3% SBA-15-amino.

The impact strengths of the glassy, as well as the tensile strengths of both the glassy and rubbery SBA-15 mesocomposites, were determined via respective dynamometry tests. The results are listed in Tables 3 and 4 for the rubbery and glassy materials, respectively. The stress–strain graphs for the rubbery mesocomposites are shown in Figure 5, and for the glassy mesocomposites, in Figure 6.

**Table 3.** Mechanical and thermomechanical properties of rubbery DEGBA/Jeffamine D-2000/SBA-15 mesocomposites.

| Mesoporous Silica Content | Tensile Strength | | | | DMA | |
|---|---|---|---|---|---|---|
| | Stress at Break (MPa) ±0.1 | Elongation at Break (%) ±3 | Modulus (MPa) ±0.5 | Toughness (kJ/m³) ±15 | Storage Modulus at Glassy State [1] (MPa) ±250 | $T_g$ [2] (°C) ±2 |
| - | 0.47 | 25.9 | 4.7 | 69.9 | 2566 | −27.7 |
| 1% SBA-15 (10) | 0.55 | 33.1 | 4.4 | 102.3 | 2808 | −31.9 |
| 3% SBA-15 (10) | 0.76 | 29.4 | 6.2 | 121.8 | 3038 | −29.1 |
| 6% SBA-15 (10) | 0.97 | 31.3 | 7.9 | 166.5 | 3704 | −26.4 |
| 9% SBA-15 (10) | 1.19 | 36.6 | 8.4 | 242.6 | 3813 | −25.8 |
| 3% SBA-15 (5) | 0.72 | 43.0 | 4.7 | 176.9 | 3447 | −24.2 |
| 3% SBA-15 (sc) | 0.89 | 47.5 | 5.4 | 241.9 | 3548 | −24.6 |
| 3% SBA-15-propyl | 0.65 | 30.5 | 5.8 | 113.2 | 2360 | −24.8 |
| 3% SBA-15-epoxy | 0.86 | 33.1 | 6.4 | 155.8 | 3949 | −22.3 |
| 3% SBA-15-amino | 1.04 | 52.9 | 5.7 | 323.0 | 4274 | −25.0 |

[1] Storage modulus values were determined at −70 °C. [2] $T_g$ was determined by the peak of tanδ vs. temperature curve (not shown) for each sample.

**Table 4.** Mechanical and thermomechanical properties of glassy DEGBA/IPD/SBA-15 mesocomposites.

| Mesoporous Silica Content | Tensile Strength | | | | Izod Impact Test | DMA | |
|---|---|---|---|---|---|---|---|
| | Stress at Break | Elongation at Break | Modulus | Toughness | Impact Strength | Storage Modulus at Glassy State [1] | $T_g$ [2] |
| | (MPa) | (%) | (MPa) | (kJ/m$^3$) | (kJ/m$^2$) | (MPa) | (°C) |
| | ±5 | ±1 | ±0.5 | ±10 | ±0.3 | ±200 | ±1 |
| - | 45.1 | 5.5 | 2349 | 1427 | 0.720 | 1203 | 151.3 |
| 1% SBA-15 (10) | 54.4 | 6.2 | 2541 | 1964 | 1.109 | 1351 | 151.3 |
| 3% SBA-15 (10) | 60.4 | 6.6 | 2463 | 2286 | 1.597 | 1448 | 147.5 |
| 6% SBA-15 (10) | 39.2 | 3.3 | 2653 | 671 | 2.138 | 2064 | 143.6 |
| 9% SBA-15 (10) | 33.2 | 2.6 | 2827 | 456 | 0.941 | 2054 | 139.7 |
| 3% SBA-15 (5) | 22.3 | 1.9 | 2599 | 228 | 1.370 | 1211 | 143.9 |
| 3% SBA-15 (sc) | 37.3 | 5.4 | 2548 | 1285 | 1.940 | 1514 | 148.0 |
| 3% SBA-15-propyl | 29.5 | 2.7 | 2594 | 431 | 0.708 | 1320 | 137.0 |
| 3% SBA-15-epoxy | 35.3 | 2.5 | 3106 | 471 | 2.164 | 1465 | 136.1 |
| 3% SBA-15-amino | 44.6 | 4.0 | 2785 | 976 | 1.354 | 1607 | 150.0 |

[1] Storage modulus values were determined at 30 °C. [2] $T_g$ was determined by the peak of tanδ vs. temperature curve (not shown) for each sample.

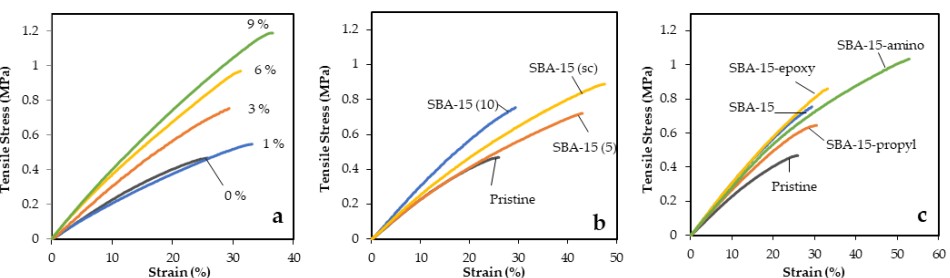

**Figure 5.** Tensile strength of rubbery epoxy mesocomposites: (**a**) effect of SBA-15 (10) concentration, (**b**) effect of SBA-15 type at 3 wt. % loading, and (**c**) effect of SBA-15 (10) organic functionalization at 3 wt. % loading.

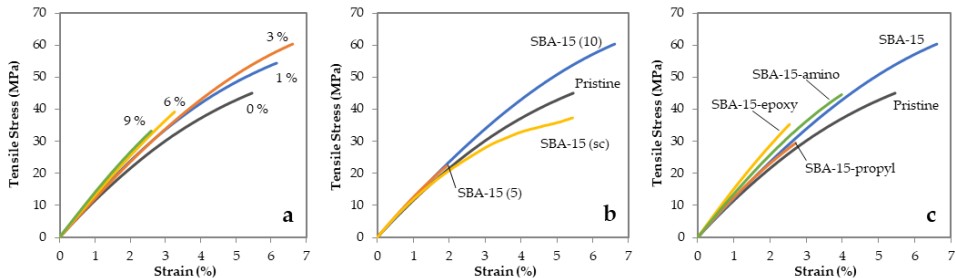

**Figure 6.** Tensile strength of glassy epoxy mesocomposites: (**a**) effect of SBA-15 (10) concentration, (**b**) effect of SBA-15 type at 3 wt. % loading, and (**c**) effect of SBA-15 (10) organic functionalization at 3 wt. % loading.

It was observed that all the rubbery SBA-15 mesocomposites were stronger and more extensible than the pristine epoxy polymers. The stress at break of the SBA-15 (10) mesocomposites increased from 0.47 to 1.19 MPa by increasing the silica content from 0 to 9%, while the elasticity of the mesocomposites was independent of the loading (Figure 5a). Among the SBA-15 additives with different porosities and morphologies (Figure 5b), the large pores and shorter SBA-15 (sc) bore the highest reinforcement by increasing both the stress and elongation at break of the pristine rubbery polymers, of up to 0.89 MPa and 47.5 %, respectively. The addition of small-pore SBA-15 (5), on the other hand, resulted in a mesocomposite with a similar strength to the SBA-15 (10) mesocomposite, but with a higher elasticity, reaching up to 43 %. Mesocomposites with functionalized SBA-15 presented superior tensile properties compared to the pristine epoxy polymer (Figure 5c). The mesocomposites of the SBA-15-epoxy and SBA-15-amino silicas with reactive groups had higher properties compared to the SBA-15 (10) mesocomposite, while the mesocomposite

of the non-reactive SBA-15-propyl had lower properties. The 1.04 MPa stress at break of the SBA-15-amino mesocomposite was the highest among the rubbery mesocomposites with organosilicas, while the 52.9% elongation at break was the highest among all the rubbery mesocomposites.

Regarding the glassy epoxy system, the reinforcement with SBA-15 mesoporous silicas resulted in mesocomposites with tensile behavior different from that of the rubbery ones. The mechanical properties of the SBA-15 (10) mesocomposites increased by increasing the silica loading up to 3 wt. %, while for higher concentrations, they decreased (Figure 6a). The mesocomposite with 3 % SBA-15 (10) presented the highest mechanical properties of all the glassy mesocomposites, with a stress at break of 60.4 MPa, elongation at break of 6.6%, tensile modulus of 2.463 GPa, and toughness of 2286 kJ/m3. The addition of SBA-15 (sc) to the glassy system resulted in a slight reduction in the tensile properties, while the addition of the small-pore SBA-15 (5) diminished the properties of the pristine epoxy polymer even further (Figure 6b). Organically modified silicas bore no improvements in the tensile properties of the glassy epoxy system at 3 % loading (Figure 6c); nonetheless, the mesocomposite with SBA-15-amino outbalanced those with SBA-15-propyl and SBA-15-epoxy.

The glassy materials were also tested for their resistance to impact. The results are listed in Table 3. The impact strengths of the mesocomposites seemed to be improved by increasing the concentration of SBA-15 (10) up to 6 %, while for 9 % loading, the resistance to impact decreased dramatically. The shorter particles of SBA-15 (sc) induced a greater improvement compared to SBA-15 (10) at the same concentration, while the smaller-pore SBA-15 (5) had a smaller effect on the impact strength. Concerning the effect of the SBA-15 functionalization, this varied among the different organic groups. The impact strength of the nSBA-15-epoxy mesocomposite was 216.4 mJ/cm$^2$, which was the highest of all the samples and practically equal to the strength of 6% SBA-15 (10). On the other hand, the improvement observed by the addition of SBA-15-amino was lower in comparison to that of SBA-15 (10), while the addition of SBA-15-propyl resulted in a decrement of the impact strength of the pristine epoxy polymer.

The viscoelastic behavior of the materials was determined via DMA. The storage modulus curves of the mesocomposites, as function of the temperature, are presented in Figures 7 and 8. The storage modulus values at the glassy state and the glass transition temperature values are shown in Table 3. In the rubbery mesocomposites, the storage modulus values at $-70\ ^\circ$C increased with SBA-15 (10) loading up to 3813 MPa, due to increased interactions of the polymer with the silica walls. The $T_g$ of the mesocomposites did not vary significantly by increasing the loading, but the differentiation became higher for the 6% SBA-15 (10) loading. The mesocomposites with SBA-15 (5) and SBA-15 (sc) had similar storage modulus values, which were higher than that of the SBA-15 (10) mesocomposite. The usage of SBA-15-bearing reactive oxirane and amine groups resulted in rubbery materials with higher storage modulus values, while the mesocomposite with a non-reactive propyl chain had a lower storage modulus than the pristine epoxy polymer. A higher increase in the $T_g$ was observed for the SBA-15-epoxy mesocomposite, which reached $-22.3\ ^\circ$C.

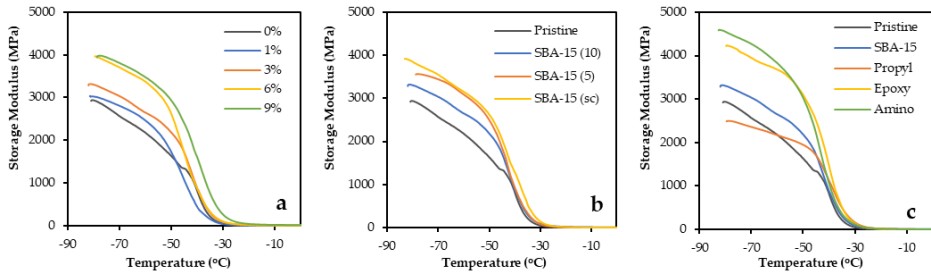

**Figure 7.** Storage modulus−temperature graphs of rubbery epoxy mesocomposites: (**a**) effect of SBA-15 (10) concentration, (**b**) effect of SBA-15 type at 3 wt. % loading, and (**c**) effect of SBA-15 (10) organic functionalization at 3 wt. % loading.

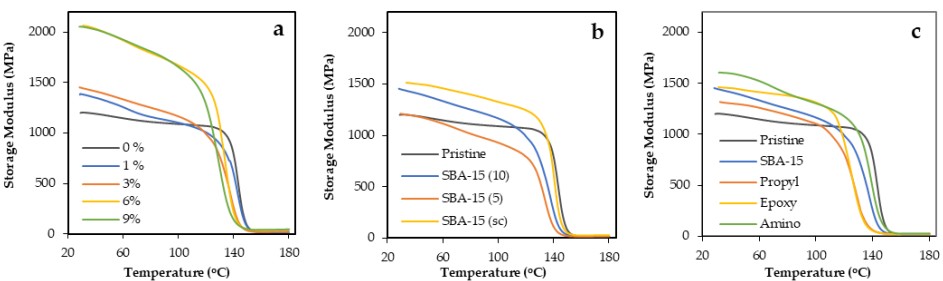

**Figure 8.** Storage modulus—temperature graphs of glassy epoxy mesocomposites: (**a**) effect of SBA-15 (10) concentration, (**b**) effect of SBA-15 type at 3 wt. % loading, and (**c**) effect of SBA-15 (10) organic functionalization at 3 wt. % loading.

With regard to the glassy samples, the storage modulus also increased by increasing silica content. However, in glassy system, the effect of the concentration on the storage modulus of the materials was higher compared to the effect of the SBA-15 type and functionalization. The mesocomposites with 6 and 9% SBA-15 (10) showed the highest storage modulus values, exceeding the pristine polymer by 71 %. The $T_g$, on the other hand decreased constantly by increasing the loading, and the $T_g$ of the 9% SBA-15 (10) sample was 139.7 °C. Among the mesocomposites with different types of SBA-15, those bearing the wider-pore SBA-15 (10) and SBA-15 (sc) had higher storage modulus values compared to the pristine polymer, while that with SBA-15 (5) had a similar storage modulus to the neat epoxy. As has been shown, the $T_g$ of the mesocomposites was affected by the pore diameters of the mesoporous silicas, but it was not affected by the particle lengths. More particularly, the $T_g$ of the mesocomposite with the small-pore SBA-15 (5) was 143.9 °C, and when the large-pore SBA-15 (10) and SBA-15 (sc) were used, the $T_g$ was 147.5 and 148.0 °C, respectively. The functionalization of the mesoporous silica resulted in an increase in the storage modulus values of the mesocomposites, but only when the organic moieties bore reactive groups. The polymers with epoxy- and amino-modified SBA-15 had higher storage modulus values than the mesocomposite with non-modified SBA-15, while the storage modulus of the SBA-15-propyl mesocomposite lay below that of the neat SBA-15 mesocomposite and above that of the pristine epoxy polymer. Unexpectedly, the same behavior was not observed for the $T_g$ of the mesocomposites with functionalized SBA-15. Although the $T_g$ of the SBA-15-amino mesocomposite was one of the highest among all the composite materials and the $T_g$ of the SBA-15-propyl was one of the lowest, the mesocomposite with the reactive SBA-15-epoxy had a $T_g$ of 136.1 °C, which was the lowest of all the mesocomposites.

The thermal stabilities of the pristine epoxy polymers and SBA-15 mesocomposites were investigated via TGA measurements and the results are summarized in Table 5. The thermal degradation profiles of both the rubbery and glassy materials were analogous up to a 50% weight loss, while further decomposition took place at higher temperatures for the glassy samples compared to the rubbery samples. More specifically, the pristine rubbery epoxy polymer lost 50 and 80% of its

mass at 388.1 and 402.6 °C, respectively, while the corresponding weight losses of the pristine glassy epoxy polymer occurred at 376.8 and 426.9 °C.

**Table 5.** Thermal decomposition temperatures of epoxy/SBA-15 mesocomposites.

| Epoxy System | Silica Filler | Temperature Values of Specific Weight Losses (%) | | | | |
|---|---|---|---|---|---|---|
| | | $T_5$ (°C) | $T_{10}$ (°C) | $T_{20}$ (°C) | $T_{50}$ (°C) | $T_{80}$ (°C) |
| Rubbery (DEGBA/ D2000) | - | 352.5 | 363.4 | 372.0 | 388.1 | 402.6 |
| | 1% SBA-15 (10) | 349.4 | 362.3 | 371.9 | 388.0 | 403.2 |
| | 3% SBA-15 (10) | 349.4 | 361.6 | 371.7 | 389.1 | 406.2 |
| | 6% SBA-15 (10) | 354.5 | 364.5 | 373.5 | 390.3 | 409.2 |
| | 9% SBA-15 (10) | 353.0 | 364.2 | 373.7 | 390.6 | 411.8 |
| | 3% SBA-15 (5) | 348.1 | 360.9 | 370.8 | 386.6 | 402.1 |
| | 3% SBA-15 (sc) | 351.4 | 363.0 | 372.4 | 388.9 | 405.4 |
| | 3% SBA-15-propyl | 353.6 | 364.0 | 373.2 | 389.1 | 405.5 |
| | 3% SBA-15-epoxy | 352.7 | 364.0 | 373.6 | 390.2 | 407.6 |
| | 3% SBA-15-amino | 347.5 | 361.1 | 370.6 | 387.7 | 403.4 |
| Glassy (DEGBA/IPD) | - | 345.5 | 354.4 | 360.6 | 376.8 | 426.9 |
| | 1% SBA-15 (10) | 347.9 | 358.0 | 364.9 | 380.7 | 445.2 |
| | 3% SBA-15 (10) | 346.4 | 353.8 | 360.6 | 380.6 | 454.8 |
| | 6% SBA-15 (10) | 345.9 | 357.7 | 365.4 | 382.2 | 491.8 |
| | 9% SBA-15 (10) | 348.4 | 356.2 | 363.3 | 384.2 | 503.7 |
| | 3% SBA-15 (5) | 346.8 | 355.0 | 362.2 | 378.7 | 435.4 |
| | 3% SBA-15 (sc) | 348.4 | 356.4 | 363.2 | 379.9 | 440.2 |
| | 3% SBA-15-propyl | 349.9 | 357.2 | 363.8 | 380.0 | 439.2 |
| | 3% SBA-15-epoxy | 346.4 | 355.6 | 363.0 | 379.6 | 436.4 |
| | 3% SBA-15-amino | 342.8 | 354.4 | 363.7 | 382.5 | 454.7 |

The most noteworthy differentiation in the thermal decomposition temperatures between the rubbery and glassy mesocomposites arose from the variability in the SBA-15 (10) concentration. As is shown in Figure 9, increasing loadings resulted in an increased thermal stability. Nevertheless, this effect was minor in the rubbery mesocomposites when compared to the glassy ones. For example, the addition of 9 % SBA-15 (10) resulted in a 9.2 °C increase in the T80 thermal decomposition temperature of the rubbery epoxy polymer, while the corresponding increase for the glassy mesocomposite was 76.8 °C. The glassy mesocomposites with SBA-15 (5) and SBA-15 (sc) were subject to an 80% mass loss at 435.4 and 440.2 °C, respectively, which were lower by 19.4 and 14.6 °C compared to the T80 of 3% SBA-15 (10). Among the glassy mesocomposites with functionalized SBA-15, the amine-modified silica induced a higher improvement in the thermal stability, which was similar to that of SBA-15 (10).

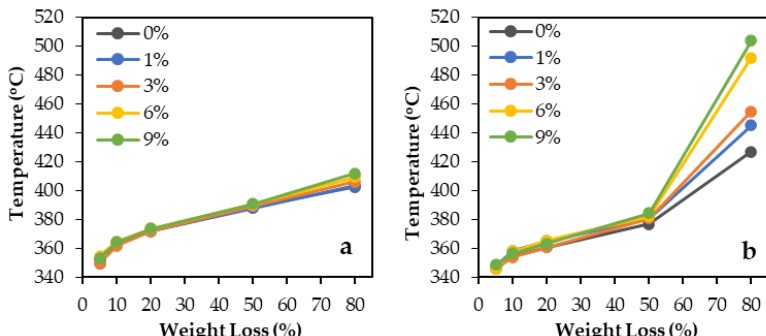

**Figure 9.** Thermal decomposition of (**a**) rubbery and (**b**) glassy epoxy mesocomposites with various loadings of SBA-15 (10).

## 4. Discussion

Before discussing the results of the polymer testing, it is essential to consider some aspects about the natures of the two epoxy systems.

- IPD is a cycloaliphatic molecule with a molar mass of 170.3 g·mol$^{-1}$ and Jeffamine D-2000 is a long-chained poly(ether) diamine with a molar mass of 2000 g·mol$^{-1}$. Thus, the curing of DGEBA with IPD results in a network with a high crosslinking density, while curing with D-2000 gives a network with a lower crosslinking density.
- The significant difference in the molar masses of the curing agents also has an impact on the mass percentages of the epoxy systems prior to curing. The percentage compositions of the masses of the rubbery and glassy systems are (i) DGEBA (26.1):D-2000 (73.9) and (ii) DGEBA (81):IPD (19), respectively. As is shown, Jeffamine D-2000 is the reagent with the higher mass fraction in the rubbery system, while DGEBA has the higher mass fraction in the glassy system.
- IPD is more reactive as a hydrogen donor than D-2000, and this may be crucial for the growth of a homogeneous crosslinked network inside the tubular SBA-15 pores.
- Considering the size of the monomers and mesoporous silicas, the molecular volume of Jeffamine D-2000 is larger by one order of magnitude than IPD and DGEBA, while a single cylindrical pore of SBA-15 is larger by four–five orders of magnitude than Jeffamine D-2000.
- For the preparation of the mesocomposites, mesoporous silicas are initially dispersed inside the epoxy resin. This means that the curing agent must penetrate inside the DGEBA-filled pores for the crosslinking network to be formed throughout the mesoporous particles.

In order for amorphous polymers bearing additives to be transparent, one of the following preconditions should be true: (i) the size of the additives must be smaller than the lower wavelength of light's visible spectrum (400 nm) or (ii) the two phases must have similar refractive index values ($n$). Since SBA-15 primary particles are bigger than 400 nm, the factor determining whether a mesocomposite is transparent or not is the matching of the refractive indices of different phases. The refractive index values of raw materials are: DGEBA ($n$ = 1.57), D-2000 ($n$ = 1.45), IPD ($n$ = 1.49), and SiO$_2$ ($n$ = 1.46). As can be seen, $n_{SiO_2}$ matches to $n_{D-2000}$ by a 0.01 variation, while the variations in $n_{SiO_2}$ between $n_{DGEBA}$ and $n_{IPD}$ are 0.11 and 0.03, respectively. By correlating the refractive index variations to the mass fractions of the components in each epoxy system, it becomes clear that the optical transparency of rubbery mesocomposites is attributed to the matching refractive index of SBA-15 to the refractive index of the reagent with the higher mass fraction, meaning the curing agent D-2000. On the other hand, the opacity of glassy mesocomposites is attributed to the mismatching of the $n_{SiO_2}$ to the $n_{DGEBA}$, which is the reagent with the higher mass fraction for this system.

The tensile properties of the epoxy polymers were generally improved by the addition of SBA-15 silicas, but the behavior of the mesocomposites under tensile stress greatly depended on the density of the crosslinked network; hence, the type of curing agent. The strengthening of the rubbery mesocomposites by increasing the silica loading, without sacrificing their elasticity, could be attributed to the extended interfacial interactions between the organic and inorganic phases [18,50]. The retaining of their elasticity could be a result of the crosslinked network growth throughout the silica particles and the orientation of the longer D-2000 segments along the tubular pores. The higher tensile properties of the SBA-15 (sc) mesocomposite derived from the higher aspect ratio of the mesoporous silica primary particles and their rod-like morphology. The Young's modulus of the rubbery mesocomposites, however, seemed to depend on the pore volume variation, rather than the pore diameter or particle morphology. The stiffness of the mesocomposites for the same additive concentration increased by increasing the SBA-15 pore volume, probably because of the larger space available to be impregnated by the polymer. The advanced tensile properties of the SBA-15-epoxy and -amino mesocomposites compared to the SBA-15-propyl one could be associated with the existence of reactive surface groups, which enabled the formation of covalent bonds between the polymer matrix and inorganic particles, thus capacitating the transfer of mechanical load from the matrix to the particles [34]. The supreme improvement induced by the functionalization of the silica with the amine groups, compared to that of

the modification with the epoxy groups, was related to the difficult impregnation of the DEGBA-filled pores by the large molecules of D-2000. The pre-existence of groups able to react with the DGEBA epoxy rings facilitated the growth of the crosslinked network deeper in the SBA-15 pores and closer to the framework walls. Concerning the glassy mesocomposites, the high density of the crosslinked network was a determinant. Although the elasticity and strength of the mesocomposites increased by increasing the SBA-15 loading up to 3 %, because of more interfacial interactions occurring, the usage of higher concentrations of mesoporous silicas caused a disturbance of the DGEBA/IPD network, leading to a decrement of the tensile properties. Nonetheless, the increasing stiffness of the mesocomposites by increasing the silica loading was due to the higher number of rigid particles inside the polymer matrix. The large pore volume of SBA-15 (10) enabled the formation of a homogeneous crosslinked network inside and outside the silica particles as well, thus permitting a homogeneous dispersion of the mechanical load throughout the epoxy matrix and facilitating a true reinforcement via the siliceous framework. As the pore volume decreased from SBA-15 (sc) to SBA-15 (5), the density of the crosslinked network inside and outside the silica particles varied further. This may be the reason for the inhomogeneous load transfer throughout the polymer matrix and the formation of areas that received higher stresses, leading to a decrement of the tensile properties. Network disturbance was also caused by the organic (reactive and non-reactive) moieties of the functionalized silicas and, as a result, the elasticity and strength of the respective meso-composites decreased. However, the covalent bonding of the epoxy polymer to the epoxy and amine groups resulted in mesocomposites with a higher Young's modulus, because of increased interfacial interactions.

The impact strength test was differentiated from the tensile strength and DMA tests by means of the applied force. In the impact tests, the force was singly applied on the specimen and then propagated through the matrix, while the applied force in the tensile and DMA tests was monotonically and periodically continuous, respectively. That said, some of the toughening mechanisms that have been proposed against impact forces are: crack pinning or deflection, additive fracture or pull-out, matrix deformation, bridging, microcracks formation, shear bending, and others [51]. With regard to the effect of the SBA-15 silicas on the resistance to impact of the glassy epoxy mesocomposites, the proportional improvement of the impact strength to the silica loading up to 6% could be attributed to the existence of more crack pinning/deflection points, but also to the decrease in the interparticle distance inside the polymer matrix [52,53]. The large decrement observed for the 9% mesocomposite, however, may have been caused by the restricted chain mobility that led to an increase in the material brittleness. The improvement induced by the SBA-15 silicas with different structural characteristics was analogous to the aspect ratio increase in the primary silica particles. SBA-15 (sc), with the higher aspect ratio, has a rod-like morphology that is known to favor impact reinforcement over spherical and platelet morphologies. Moreover, it was possible that the formation of a more extensive, 3D, crosslinked network throughout the particles was favored by the bigger diameters of the SBA-15 (sc) and SBA-15 (10) pores and resulted in an improved resistance to fracture by impact. The highest impact strength was observed for the SBA-15-epoxy mesocomposite. This result was a combination of the effective load transfer from the polymer matrix to the inorganic particles through the covalent bonds and the relatively large available (after the organic modification) pore volume for the formation of the 3D epoxy network. Unexpectedly, the impact strength of the SBA-15-amino mesocomposite was slightly lower compared to the SBA-15 (10) mesocomposite, even though the tensile and thermomechanical properties were among the best. This could be explained by the smaller pore volume/diameter available after the amination of SBA-15 (10) and the high content of the reactive amine groups that may have decreased the resistance to fracture by impact. The low impact strength of the SBA-15-propyl mesocomposite could be attributed to the existence of non-reactive propyl groups that acted as a barrier between the epoxy network and the silica surface, preventing extensive interactions.

The DMA measurements revealed that the thermomechanical properties of the mesocomposites changed individually due to the SBA-15 loading, type, and surface functionalization for each epoxy system, and that they depended greatly on the crosslinked network density. The storage modulus improvement by increasing the silica loading for both the rubbery and glassy mesocomposites could be attributed to the increased energy needed to deform the specimens, due to the higher interfacial interactions between the polymer and rigid SBA-15 particles. The changes in the mesocomposites stiffness due to the addition of SBA-15 silicas with varying pore sizes and particle size/morphologies could be directly correlated to the density of the crosslinked network. The epoxy polymer formed by the high MW, linear D-2000 curing agent had a low network density and high free volume; thus, it could be reinforced more effectively by the smaller particles of SBA-15 (5) and SBA-15 (sc). On the other hand, the dense network of DGEBA/IPD grew more extensively inside the larger pores of SBA-15 (sc) and SBA-15 (10), resulting in mesocomposites with higher storage modulus values. The existence of reactive epoxy- and amine-groups on the surface of SBA-15 enhanced the storage modulus improvement in both the rubbery and glassy samples due to increased interfacial interactions between the phases. However, the organic moieties of the SBA-15-propyl, which bore no reactive groups, lay as an intermediate layer between the epoxy matrix and siliceous framework, hindering the interfacial interactions and adversely affecting the storage modulus values of both the rubbery and glassy mesocomposites. The $T_g$ of the SBA-15 mesocomposites depended on the mobility of the polymeric chains, and the latter was individually affected by the addition of mesoporous silicas based on the type of the curing agents used. The increase in the $T_g$ due to the silica loading in the rubbery mesocomposites could be attributed to the decreased interparticle distance in the polymer matrix that restricted the chain movement in the bulk region. Although the $T_g$ at 1 and 3% SBA-15 concentrations was lower compared to the pristine polymer, due to disorders in the crosslinked network, at higher silica concentrations, the free volume of the rubbery mesocomposites decreased and the chain movements were restricted, resulting in a higher $T_g$. In the glassy mesocomposites, on the other hand, the $T_g$ was constantly decreased by the silica loading because of the increased network distortion, which could not be counterbalanced by restrictions of the chain movements in the bulk matrix, due to the high network density of the DEGBA/IPD system. The differences in the pore volumes among SBA-15 (10), SBA-15 (5), and SBA-15 (sc) caused different changes in the $T_g$ of the rubbery and glassy materials, which could be correlated with the molecular size of the curing agent. The rubbery mesocomposites with SBA-15 (5) and SBA-15 (sc) had higher $T_g$ due to the smaller pores of the additives, which caused bigger restrictions to the movement of the large D-2000 molecules. On the contrary, the SBA-15 (sc) and SBA-15 (10) had pores with larger diameters and allowed the growth of a more extensive DEGBA/IPD network, which resulted in a higher $T_g$ compared to the SBA-15 (5) glassy mesocomposite. The effect of SBA-15 functionalization on the mesocomposites' $T_g$ could be related to the activity of each curing agent, as well as to the steric obstructions caused by the organic moieties inside the SBA-15 pores. The presence of organic groups favored the $T_g$ increase in the rubbery mesocomposites over the pristine polymer and the mesocomposite with unmodified SBA-15, because of restrictions in the chain movements inside the silica pores. Furthermore, the lower $T_g$ of the SBA-15-amino mesocomposite, compared to that of the SBA-15-epoxy, could be attributed to the lower pore volume of the amine-modified silica, combined with the higher concentration of organic groups that may have hindered the pore impregnation with the large D-2000 molecules. The high $T_g$ of the glassy SBA-15-amino mesocomposite resulted from the growth of a robust epoxy network inside the silica pores and the large number of covalent bonds formed between the polymer matrix and amine groups of SBA-15. The low $T_g$ of the mesocomposites with propyl- and epoxy-modified SAB-15 could be attributed to the existence of free organic chains in the polymer mass that favored the viscous dissipation of energy as heat at lower temperatures. This was something to be expected from the non-reactive propyl groups; however, the lack of reaction for the surface epoxy groups of the, respectively, modified SBA-15 could be explained by

the higher reactivity of the IPD, since the cycloaliphatic curing agent may have reacted first with the DGEBA molecules inside the silica pore, creating a barrier in front of the inner oxirane rings of the SBA-15-epoxy; thus, the viscosity of these groups increased in lower temperatures.

The higher resistance to the thermal degradation of the glassy epoxy system could be attributed to the higher crosslinking density [54,55]. The increase in the thermal stability by increasing the mesoporous silica content was the outcome of the larger number of pores available to the host polymer chains, which restricted their movement. The great divergence in the thermal stability of the two epoxy systems by increasing the SBA-15 loading was attributed to the different sizes of the curing agents. A larger number of the smaller IPD molecules could enter the SBA-15 pores more easily and form a crosslinked network with a higher density. The "protecting" effect of the siliceous pores was also evidenced by the fact that the glassy SBA-15 (10) mesocomposite had the highest thermal stability compared to the SBA-15 (5) and SBA-15 (sc) ones, because of the larger pore size of SBA-15 (10) and its ability to host greater amounts of crosslinked polymers. The higher thermal stability of the mesocomposites with SBA-15-amino was attributed to increased interactions between the silica and polymer that, subsequently, could be correlated with the higher organic group content, as well as with the preparation procedure of the mesocomposite, where the additive was initially dispersed inside the epoxy resin and the bonding between the amine groups of silica and the oxirane rings was favored.

## 5. Conclusions

In the present work, glassy and rubbery epoxy/SBA-15 mesocomposites were successfully prepared. The mesoporous silica particles were homogeneously dispersed inside the polymer matrix and the effects of the silica concentration, porosity, particle morphology, and surface functionalization were studied.

The glassy epoxy/SBA-15 composites were all opaque due to the refractive index mismatching between the silica particles and the DEGBA epoxy resin. Their mechanical and viscoelastic properties were improved with up to a 6 wt. % addition of SBA-15 silica, while the thermal properties of the composites were not affected. SBA-15 variants with higher surface areas and pore volumes induced a more pronounced improvement in all the (thermo)mechanical properties. Finally, the functionalization of the SBA-15 surface with organic molecules (amine and epoxy groups) further improved the properties of yhe epoxy composites.

Regarding the rubbery epoxy/SBA-15 composites, they were all transparent due to the refractive index matching between the silica particles and the Jeffamine D-2000 curing agent, which was in abundance compared to the DEGBA prepolymer. All the studied properties were improved by increasing the SBA-15 loading up to 9 %, while the porosity characteristics (surface area and pore volume), particle size, and aspect ratio had different effects on the (thermo)mechanical properties of the composites. A similar beneficial effect of the SBA-15 functionalization was found for the glassy composites.

The observed improvement in the properties of both the glassy and rubbery epoxy polymers due to the addition of mesoporous silicas offers further opportunities for the existing wide range of applications of epoxy polymers, including their use as adhesives, coatings, structural and construction parts, protection/restoration of marbles, automotive, aerospace, shipbuilding, sports, electronics, and many more.

**Author Contributions:** Conceptualization, D.G. and K.T.; methodology, D.G., D.B., D.E. and K.T.; formal analysis, D.G.; investigation, D.G.; resources, D.B. and K.T.; writing—original draft preparation, D.G.; writing—review and editing, D.G. and K.T.; supervision, K.T.; funding acquisition, D.E. and K.T. All authors have read and agreed to the published version of the manuscript.

**Funding:** This research has been co-financed by the European Regional Development Fund of the European Union and Greek national funds through the Operational Program Competitiveness,

Entrepreneurship and Innovation (EPAnEK 2014–2020), under the Action "RESEARCH—CREATE—INNOVATE B' CALL" (project code: T2EDK-02205).

**Institutional Review Board Statement:** Not applicable.

**Informed Consent Statement:** Not applicable.

**Data Availability Statement:** Not applicable.

**Conflicts of Interest:** The authors declare no conflict of interest.

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
