# Peer review of "Glassy and Rubbery Epoxy Composites with Mesoporous Silica"

_jcs, doi:10.3390/jcs7060243_

Round 1
Reviewer 1 Report
Recommendation: Accept after minor revision
The manuscript “Glassy and rubbery epoxy composites with mesoporous silica” aimed to provide insights into the effect of various characteristic silica nanoparticles (NPs) on the thermal and mechanical properties of epoxy. The authors synthesized three silica NPs with different pore sizes and aspect ratio as the filler of interests. Moreover, the NPs were functionalized with three different organic molecules to probe the effect of interfacial properties. Both glassy and rubbery epoxy were synthesized and compared to provide a better understanding of the role of silica NPs on polymers with drastically different mechanical properties. The authors provided a thorough discussion on the subject and the data, and touched almost every aspect of polymer nanocomposite (i.e., particle dimension, pore environment, cross-linking, particle-polymer interface).
Major comments:
1. The authors discussed morphologies of agglomerates of three types of fillers based on SEM (section 3.1). However, the morphologies shown in SEM do not represent the actual morphologies of particles because the drop-cast particle suspension slowly evaporates on SEM stage and can induce aggregation. When particles are dispersed in solvent or polymer matrix the aggregation behavior is most likely different from the SEM images. SEM can only be used to observe particle shape and size. In order to comment on agglomeration state, in-situ techniques such as DLS are needed. The authors should re-phrase these statements.
2. Following above point, line 233-234 stated that the SBA-15(10) is well-dispersed within the epoxy and “disaggregated”. As mentioned above, it is better to rephrase. More importantly, I recommend using some type of “bulk” measurement such as SAXS to quantify the particle dispersion, or even just optical microscopy. Given the particle size (~200 nm), optical microscope should be able to tell the particle dispersion. One or two TEM images are not convincing enough to state that the particles are well-dispersed.
3. The authors did an excellent job showing extensive results and discussion on thermal-mechanical properties of the nanocomposites. Although I agree with showing representative curve of tensile test, TGA, DMA etc., I strongly recommend providing error bars on the obtained results, including Young’s modulus, storage modulus, tensile strength, degradation temperature, particle dimensions (length, aspect ratio etc.) and other thermal/mechanical properties. One single value is not scientifically sufficient to draw conclusion.
Minor comments:
1. Line 50, the authors introduced using silica NPs for mechanical reinforcement. But how strong is silica? What are its mechanical properties? It would be nice to have this information here.
2. Can the authors show TEM of all three silica NPs in Figure 2? TEM has higher resolution and can help determine aspect ratios better than SEM. In fact, the SEM images (Figure 1b, d, and e) are blurry.
2. The authors stated that the decrease of surface areas and pore sizes proved the successful incorporation of organic modifiers. It is important to provide other more direct evidence to quantify this statement. I would suggest using NMR, FTIR, and XPS on pristine and functionalized silica NPs.
Author Response
Major comments:
- The authors discussed morphologies of agglomerates of three types of fillers based on SEM (section 3.1). However, the morphologies shown in SEM do not represent the actual morphologies of particles because the drop-cast particle suspension slowly evaporates on SEM stage and can induce aggregation. When particles are dispersed in solvent or polymer matrix the aggregation behavior is most likely different from the SEM images. SEM can only be used to observe particle shape and size. In order to comment on agglomeration state, in-situ techniques such as DLS are needed. The authors should re-phrase these statements.
Response: We thank the reviewer for the insightful comment. Indeed, drop casting/evaporation method may induce “pseudo-aggregation”. However, in our case, the SEM samples were prepared by simply mounting small amount of powder on the SEM holder followed by gentle shaking to remove easily detachable dust/powder. This detail has been added in the experimental section. Furthermore, according to the reviewer’s comment, we have also added a note, indicating that the observed aggregated clusters in SEM images may partially disaggregate when the meso-silica is dispersed in the epoxy (pre)polymer, as is also revealed by the TEM images of the composites.
- Following above point, line 233-234 stated that the SBA-15(10) is well-dispersed within the epoxy and “disaggregated”. As mentioned above, it is better to rephrase. More importantly, I recommend using some type of “bulk” measurement such as SAXS to quantify the particle dispersion, or even just optical microscopy. Given the particle size (~200 nm), optical microscope should be able to tell the particle dispersion. One or two TEM images are not convincing enough to state that the particles are well-dispersed.
Response: Following up on the previous response, the phrase “disaggregated” was replaced by “partially disaggregated” on the basis of the through inspection of the TEM specimens, which verified that the long/big aggregates of particles observed in the SEM images were not predominant within the bulk epoxy polymer. Certainly, few aggregated particles were also observed. Optical microscopy was not of big help, we will perform SAXS measurements in the future although we are not sure they could be also informative as could be in the case of order nano-layered fillers.
- The authors did an excellent job showing extensive results and discussion on thermal-mechanical properties of the nanocomposites. Although I agree with showing representative curve of tensile test, TGA, DMA etc., I strongly recommend providing error bars on the obtained results, including Young’s modulus, storage modulus, tensile strength, degradation temperature, particle dimensions (length, aspect ratio etc.) and other thermal/mechanical properties. One single value is not scientifically sufficient to draw conclusion.
Response: Standard error values have been added to the tables of mechanical and thermomechanical properties. The initial table had to be split into two separate tables, since the standard error values were different between glassy and rubbery materials. Concerning the thermal decomposition temperatures, standard error values were not added in the respective Table, since each sample was tested once.
Minor comments:
- Line 50, the authors introduced using silica NPs for mechanical reinforcement. But how strong is silica? What are its mechanical properties? It would be nice to have this information here.
Response: We thank the reviewer for the apt comment. Although most of the mechanical properties of silicon dioxide particles are well documented, in the case of mesoporous silica the mechanical properties are determined mainly when the material is produced in the form of thin film. Since mesoporous silica is used in the form of dispersed particles in our study, the information for the mechanical properties of crosslinked network of more classical silicon dioxide was added at the point indicated by the reviewer.
- Can the authors show TEM of all three silica NPs in Figure 2? TEM has higher resolution and can help determine aspect ratios better than SEM. In fact, the SEM images (Figure 1b, d, and e) are blurry.
Response: A TEM image of SBA-15 (10) was added in Figure 2. Unfortunately, there are no TEM images of SBA-15 (5). Furthermore, we have added more high-resolution TEM images of the epoxy – silica composites, in which the dispersion of the silica particle within the bulk epoxy polymer can be clearly seen. The SEM and TEM images that were selected were the most representative and of the best quality compared to the sum of the available images. Trying our best, the images were post-treated to increase their clarity and contrast.
- The authors stated that the decrease of surface areas and pore sizes proved the successful incorporation of organic modifiers. It is important to provide other more direct evidence to quantify this statement. I would suggest using NMR, FTIR, and XPS on pristine and functionalized silica NPs.
Response: The quantification of the organo-functionalization of the parent SBA-15 silica was based on elemental carbon analysis measurements and these data were discussed more clearly in the revised manuscript. The careful analysis and interpretation of the N2 sorption (surface area, pore width and volume) allowed for further insight, such as that the organic groups have been inserted within the pores and not just blocking the windows/opening of the pores. Furthermore, it is evident that the higher the loading of the organic group, the more pronounced the reduction of surface area and pore volume, almost by 50%, without however a significant narrowing of the pores, which would lead to inhibition of the interaction with the epoxy prepolymer. The above suggested characterization techniques would certainly offer complementary information regarding the successful functionalization and are planned as future work.
Reviewer 2 Report
This work is interesting but here mentioning a few comments to improve the journal's quality.
1. Suggesting authors revise the title " Glassy and rubbery epoxy composites with mesoporous silica" of this work for better reach.
2. Similar material composites were reported in different methods and approaches, its better to tabulate and compare for quick check and reference.
3. Conclusion section should be revised for a concise version.
4. Can authors elaborate on synthesized "glassy and rubbery epoxy composites with mesoporous silica" material applications?
5. The figure's quality is poor need to revise them with good-quality images.
Author Response
This work is interesting but here mentioning a few comments to improve the journal's quality.
- Suggesting authors revise the title " Glassy and rubbery epoxy composites with mesoporous silica" of this work for better reach.
Response: We thank the reviewer for his/her suggestion; however, the title was selected as it is generic enough to attract increased interest, and at the same time not to exaggerate with regard to the actual content of the paper.
- Similar material composites were reported in different methods and approaches, its better to tabulate and compare for quick check and reference.
Response: In the third paragraph of the Introduction, a wide variety of polymers and mesoporous silica (nano)composites were discussed, supported by more than 20 references. Effects on physiochemical characteristics and performance properties owning to mesoporous silica addition was identified, with emphasis on the benefits induced from the high surface area of the mesoporous silica and the enhanced interfacial interactions with the (pre)polymers.
- Conclusion section should be revised for a concise version.
Response: The Conclusions was reduced substantially, focusing on the most important findings that have derived by combining the properties of the SBA-15 mesoporous particles with the glassy and rubbery epoxy polymers.
- Can authors elaborate on synthesized "glassy and rubbery epoxy composites with mesoporous silica" material applications?
Response: A relevant comment has been added in the conclusions, in order to highlight the potential of these epoxy composites on the basis of the improved performance properties obtained by the addition of the mesoporous silica.
- The figure's quality is poor need to revise them with good-quality images.
Response: We have revised the TEM images and carefully checked/updated the rest of the figures.
Round 2
Reviewer 1 Report
Great updates from the authors. All of my comments were properly addressed and the manuscript is recommended for publication.
Author Response
We thank the reviewer for his/her response.
Sugested by the Reviewer 2, the manuscript has been edited accordigly for the colnclusions section to be more concise.
Reviewer 2 Report
The authors revised the manuscript and made all the required changes to the manuscript. However, suggesting to edit the conclusion section and make it concise for quick reference/check.
Author Response
We thank the reviewer for his/her response. The manuscript has been edited accordigly for the colnclusions section to be more concise.
Round 3
Reviewer 2 Report
The authors revised the manuscript with all the required edits, but the authors uploaded the final version with all the markups. Which made it difficult to read in between content. Please pay attention before submission.
But the revised manuscript significantly improved.